# Higher temperature accelerates the aging-dependent weakening of the melanization immune response in mosquitoes

**Lindsay E. Martin, Julián F. Hillyer** *

Department of Biological Sciences, Vanderbilt University, Nashville, Tennessee, United States of America

* julian.hillyer@vanderbilt.edu

**Data Availability Statement:** All relevant data are within the manuscript and its Supporting Information files.

## Abstract

The body temperature of mosquitoes, like most insects, is dictated by the environmental temperature. Climate change is increasing the body temperature of insects and thereby altering physiological processes such as immune proficiency. Aging also alters insect physiology, resulting in the weakening of the immune system in a process called senescence. Although both temperature and aging independently affect the immune system, it is unknown whether temperature alters the rate of immune senescence. Here, we evaluated the independent and combined effects of temperature (27˚C, 30˚C and 32˚C) and aging (1, 5, 10 and 15 days old) on the melanization immune response of the adult female mosquito, *Anopheles gambiae*. Using a spectrophotometric assay that measures phenoloxidase activity (a rate limiting enzyme) in hemolymph, and therefore, the melanization potential of the mosquito, we discovered that the strength of melanization decreases with higher temperature, aging, and infection. Moreover, when the temperature is higher, the aging-dependent decline in melanization begins at a younger age. Using an optical assay that measures melanin deposition on the abdominal wall and in the periostial regions of the heart, we found that melanin is deposited after infection, that this deposition decreases with aging, and that this aging-dependent decline is accelerated by higher temperature. This study demonstrates that higher temperature accelerates immune senescence in mosquitoes, with higher temperature uncoupling physiological age from chronological age. These findings highlight the importance of investigating the consequences of climate change on how disease transmission by mosquitoes is affected by aging.

## Author summary

Climate change is increasing global temperatures, and this is altering the ecosystems that organisms inhabit. Insects are particularly susceptible to temperature changes because their body temperature is dictated by the environmental temperature in which they reside. Importantly, changes in temperature alter how their immune system combats infection. Another factor that alters immunity is aging. Specifically, aging weakens immune proficiency in a process called senescence. Although both temperature and aging affect the

**Funding:** This work was funded by National Science Foundation (NSF) Grant IOS-1936843 to JFH and NSF Graduate Research Fellowship to LEM. The funders had no role in study design, data collection and analysis, decision to publish, or preparation of the manuscript.

**Competing interests:** The authors have declared that no competing interests exist.

immune system, an unanswered question is whether temperature alters the progression of immune senescence. We tested this question in *Anopheles gambiae*, which is a major vector of malaria, by measuring the strength of the melanization immune response of mosquitoes reared at three different temperatures, at four different ages, under four different infection conditions. We determined that, individually, higher temperature and aging both weaken melanization, and importantly, that higher temperature accelerates the senescence of the melanization immune response. These findings demonstrate a real-life consequence of climate change on insect physiology and illustrate the need to holistically scrutinize the impact of global warming on the ability of insects to transmit diseases to humans, animals and plants, and on the ability of insects to serve as pollinators for our food supply.

## Introduction

Mosquitoes, like most insects, are both ectotherms and poikilotherms, meaning that their body temperature is dictated by the environmental temperature in which they reside. Climate change is causing an increase in environmental temperatures, and global temperatures are predicted to further rise by more than 1.5°C between 2030 and 2052 [1]. Increasing environmental temperature is, in turn, raising the body temperature of insects, and this phenomenon is more pronounced for species that inhabit the tropics [2–4]. This is particularly the case for mosquitoes that transmit disease [5,6]; at higher temperatures the rate of development is faster [7–10], the metabolic rate is higher [2,11], the body size is smaller [12,13], and the lifespan is shorter [7,9,14].

Changes in environmental temperature also affect the immune system of mosquitoes [15–17]. Higher temperature reduces the phagocytic activity of hemocytes, which are mosquito immune cells [15]. Higher temperature also weakens melanization [15,18–20], which is a humoral immune response that encases and kills bacteria, malaria parasites, fungi, viruses, and filarial worms [20–25]. In contrast, higher temperature increases the transcription and activity of nitric oxide synthase [15,26], which produces the antibacterial and antimalarial free radical, nitric oxide [27,28]. Additionally, changes in temperature alter the expression of genes that encode components of the Toll pathway, apoptosis pathways, antimicrobial peptides, and other immune factors [15,19,29]. Because of these and other changes, the environmental temperature affects the probability that a mosquito transmits disease [16,19,30–33].

Another factor that affects the immune system of mosquitoes is aging. Mosquitoes, like most animals, undergo senescence, which is the gradual and irreversible deterioration of the efficiency of physiological processes that occurs with aging [34–40]. Senescence includes an aging-associated decline in immunity, leading to increased pathogen proliferation and increased risk of mortality [41–47]. For example, aging weakens the melanization immune response [47–52], and decreases the number of hemocytes available to quell an infection [42,48,53,54].

Although the independent effects of temperature and aging on the immune system have been investigated, it is unknown how temperature and age interact to shape the strength of the immune response of a mosquito or any other insect. That is, we do not know whether the environmental temperature influences the rate of immune senescence. We hypothesize that higher temperature uncouples physiological age from chronological age, thereby accelerating the progression of immune senescence. To test this hypothesis, we evaluated the independent and combined effects of temperature and aging on the melanization immune response of the mosquito, *Anopheles gambiae*, which is a major vector of malaria in sub-Saharan Africa (Fig 1)

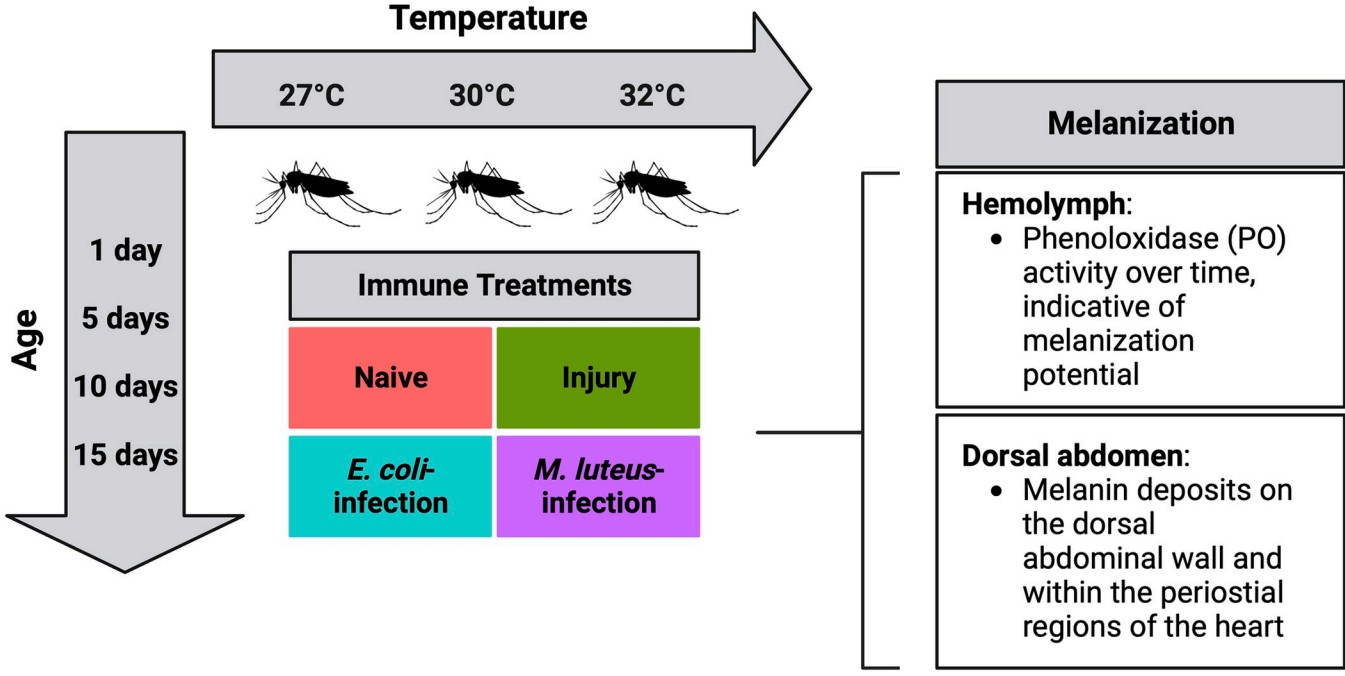

**Fig 1. Experimental overview for investigating the effects of higher temperature, aging, and their interaction on the mosquito melanization immune response.** Figure created with BioRender.com.

[55]. We found that, individually, higher temperature and aging weaken the melanization immune response, and that higher temperature accelerates the senescence of melanization.

## Results

### Melanization potential decreases with aging and higher temperature, and the aging-dependent decline is accelerated by higher temperature

Phenoloxidase (PO) initiates the melanization immune response by catalyzing the hydroxylation of tyrosine to dopa and the oxidation of dopa to dopaquinone, which then spontaneously converts to the melanin precursor, dopachrome [56, 57]. To test the effects of higher temperature and aging on the melanization potential of hemolymph, we isolated hemolymph, incubated it with the PO substrate, L-DOPA, and measured $OD_{490}$ 30 min later (Fig 2). This assay captures melanization potential because it measures the ability of hemolymph to melanize in the event that the phenoloxidase-based cascade is turned on to its fullest. A higher $OD_{490}$ indicates a higher melanization potential, or a greater ability to melanize a pathogen upon infection.

Mosquitoes at higher temperatures have a lower melanization potential regardless of age or their infection status, with temperature accounting for 31% of the variation (Fig 3A and 3D). Relative to mosquitoes at 27°C, the melanization potential of mosquitoes at 30°C and 32°C was 54% and 79% lower, respectively.

Aging reduces melanization potential regardless of temperature or infection status, with age accounting for 10% of the variation (Fig 3B and 3D). Relative to 1-day-old mosquitoes, the melanization potential of 5-day-old and 10-day-old mosquitoes was 36% and 52% lower, respectively. The melanization potential of 15-day-old mosquitoes was 58% lower than 1-day-old mosquitoes, indicating that aging beyond 10 days does not further reduce melanization potential.

Infection for 24 h reduces the melanization potential of the hemolymph regardless of temperature or age, with infection status accounting for 13% of the variation (Fig 3C and 3D).

## A. Melanization cascade

## B. Phenoloxidase assay

## C. Melanin deposition

**Fig 2. Diagrammatic overview of the melanization biochemical cascade and the experimental workflow. A.** Melanization biochemical cascade. **B.** Workflow of the phenoloxidase spectrophotometric assay that measures melanization potential of the hemolymph and PO activity over time. **C.** Representative images of melanization on the dorsal abdominal wall of 1-day-old mosquitoes reared at 27˚C, at 24 h following treatment. Melanin deposition results in dark deposits. Panel B created with BioRender.com.

Relative to naïve mosquitoes, the melanization potential was 62% and 36% lower in *E. coli*- and *M. luteus*-infected mosquitoes, respectively, and injury also reduced the melanization potential by 22%. The large reduction in melanization potential during an infection indicates that PO is being depleted as it is used to fight the infection, and is an agreement with the findings in an earlier study [51]. Moreover, the trend for lower melanization potential in injured mosquitoes can be attributed to PO being used to heal the wound [58].

Temperature and age interact to reduce the melanization potential, with this interaction accounting for 5% of the variation (Figs 3D and 4). Specifically, higher temperature accelerates the aging-dependent decline in melanization potential. As an example using naïve mosquitoes only, and relative to 1-day-olds maintained at 27˚C, 1-day-olds maintained at 32˚C had a melanization potential that was 80% lower. As mosquitoes aged from 1 to 10 days old, the melanization potential of mosquitoes at 27˚C decreased by 52%, but the melanization potential of mosquitoes at 32˚C remained the same. As another example using *M. luteus*-infected mosquitoes only, and relative to 1-day-olds maintained at 27˚C, 1-day-olds maintained at 32˚C had a melanization potential that was 82% lower. As mosquitoes aged from 1 to 10 days old, the melanization potential of mosquitoes at 27˚C decreased by 53%, but the melanization potential of mosquitoes at 32˚C remained the same. This signifies that at higher temperatures, the decrease in melanization potential occurs earlier in life than at cooler temperatures. Other interactions, such as the interaction between temperature and immune treatment, did not meaningfully shape melanization potential, so they were excluded from the model-of-best-fit.

To confirm that the change in $OD_{490}$ was due to PO activity and not the auto-oxidation of L-DOPA, every biological trial included a control composed of only L-DOPA and water. Auto-oxidation of L-DOPA was not detected (S1 Fig). Furthermore, to determine whether the change in $OD_{490}$ was enhanced by endogenous substrates in the hemolymph and not just the exogenous L-DOPA that was added to the hemolymph, a subset of hemolymph samples was

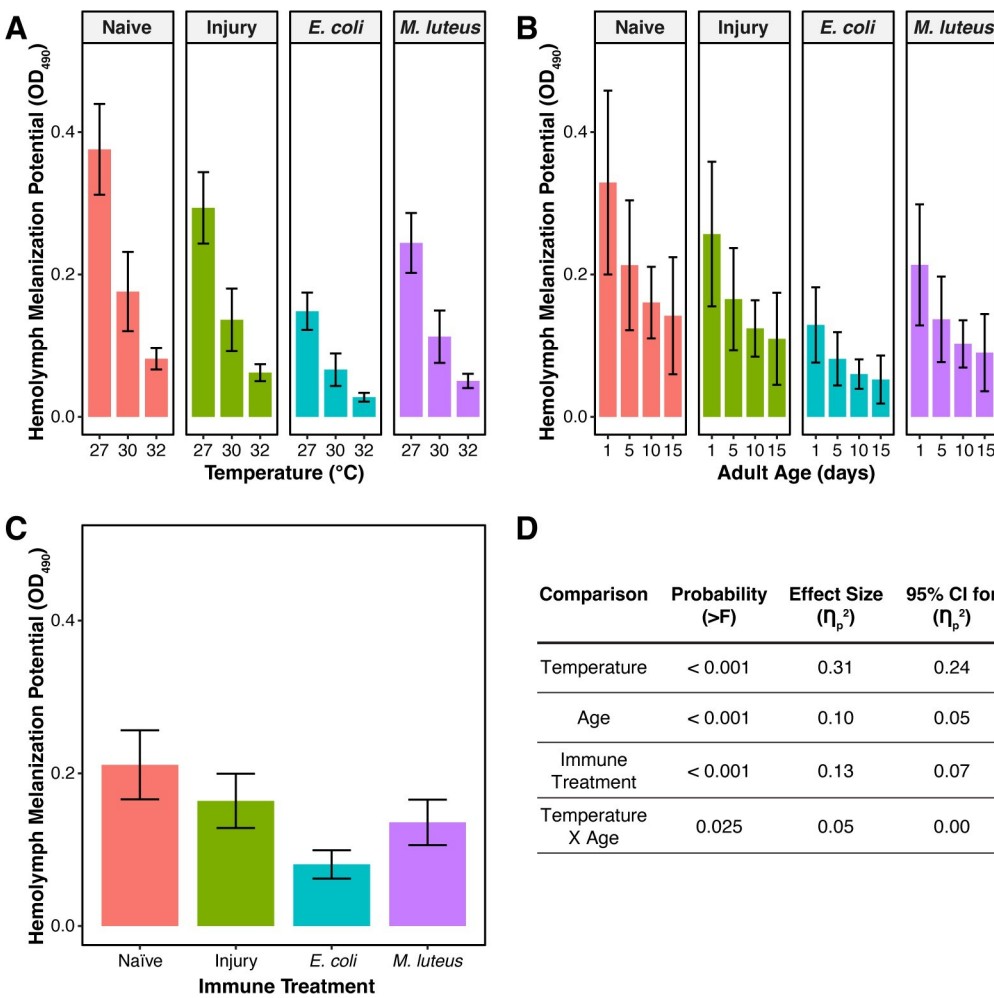

**Fig 3. Melanization potential decreases with higher temperature, aging, and infection. A.** Melanization potential, aggregated by temperature and immune treatment, irrespective of age. **B.** Melanization potential, aggregated by age and immune treatment, irrespective of temperature. **C.** Melanization potential, aggregated by immune treatment, irrespective of temperature or age. Column height marks the estimated marginal mean, and whiskers indicate the standard error of the estimated marginal mean (S.E.M.). **D.** Statistical analyses of the data using a linear model and a Type II ANOVA. The same measurements are plotted in Figs 3 and 4, but grouped or arranged differently, with aggregated data shown in this figure.

mixed with water that was devoid of exogenous L-DOPA. Without exogenous L-DOPA, a change in $OD_{490}$ was not detected (S2 Fig).

In summary, the melanization potential of hemolymph significantly decreases with higher temperature, aging, and infection. Furthermore, when the temperature is higher, the aging-dependent decline in melanization begins at a younger age, and therefore, we conclude that higher temperature accelerates the senescence-based deterioration of melanization potential.

### The speed of melanization decreases with aging and higher temperature, and the aging-dependent decline is accelerated by higher temperature

Because the melanization potential described above only captures PO activity at a single point in time, we next analyzed how higher temperature, aging, and their interaction alter the speed

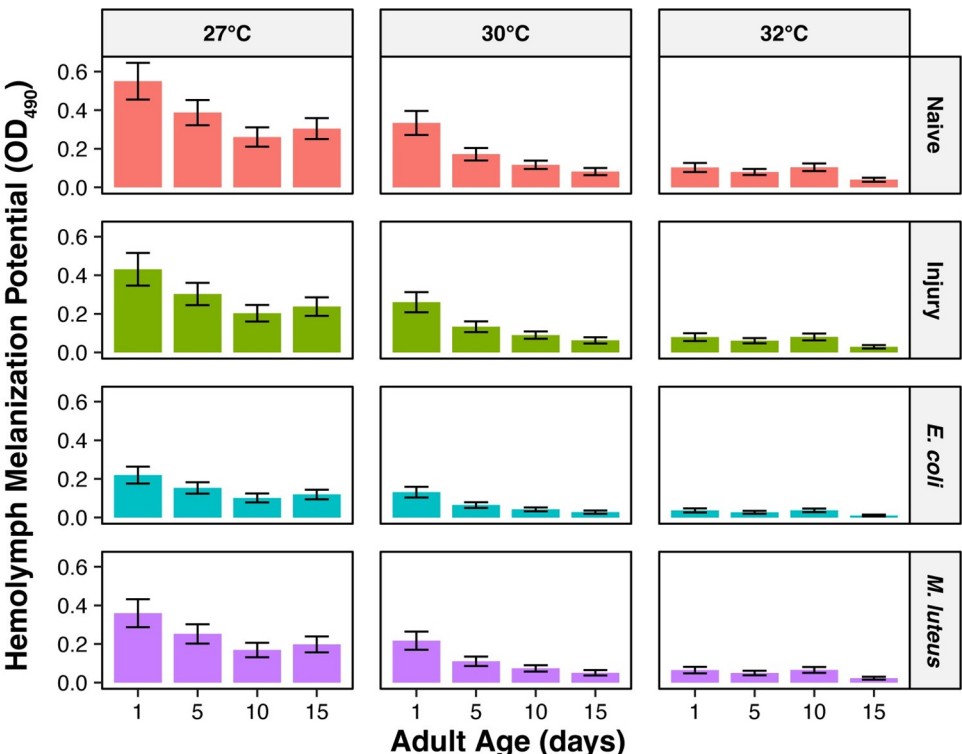

**Fig 4. The aging-dependent decline in melanization is accelerated by higher temperature.** Column height marks the estimated marginal mean, and whiskers indicate the S.E.M. The same measurements are shown in Figs 3 and 4, but grouped or arranged differently, with unaggregated data shown in this figure. The outcomes of statistical analyses are presented in Fig 3D.

of melanization over the time course of the 30 min assay. Similar to what we found for the overall melanization potential, higher temperature reduces the speed of melanization. At every timepoint in the assay, and regardless of age or infection status, mosquitoes at higher temperatures had a slower pace of melanization, with temperature accounting for 25% of the variation (Fig 5A and 5D). Combining the timepoint values for each individual temperature, and relative to mosquitoes at 27°C, the PO activity of mosquitoes at 30°C and 32°C was 56% and 76% lower, respectively.

Aging also reduces the speed of melanization regardless of temperature or infection status, with age accounting for 11% of the variation (Fig 5B and 5D). The speed of melanization was fastest in 1-day-old mosquitoes, and while the endpoint of PO activity (the melanization potential at 30 min) was similar for 10- and 15-day-old mosquitoes, the oldest mosquitoes reached their maximum PO activity later in the assay. Combining the timepoint values for each individual age, and relative to 1-day-old mosquitoes, the PO activity of mosquitoes that were 5, 10, and 15 days old was 32%, 53%, and 57% lower, respectively.

Infection for 24 h reduces the speed of melanization regardless of temperature or age, accounting for 12% of the variation (Fig 5C and 5D). Combining the timepoint values for each individual immune treatment, and relative to naïve mosquitoes, the PO activity of hemolymph was 58% and 40% lower in *E. coli*-infected and *M. luteus*-infected mosquitoes, respectively, whereas the PO activity of hemolymph was only 11% lower in injured mosquitoes.

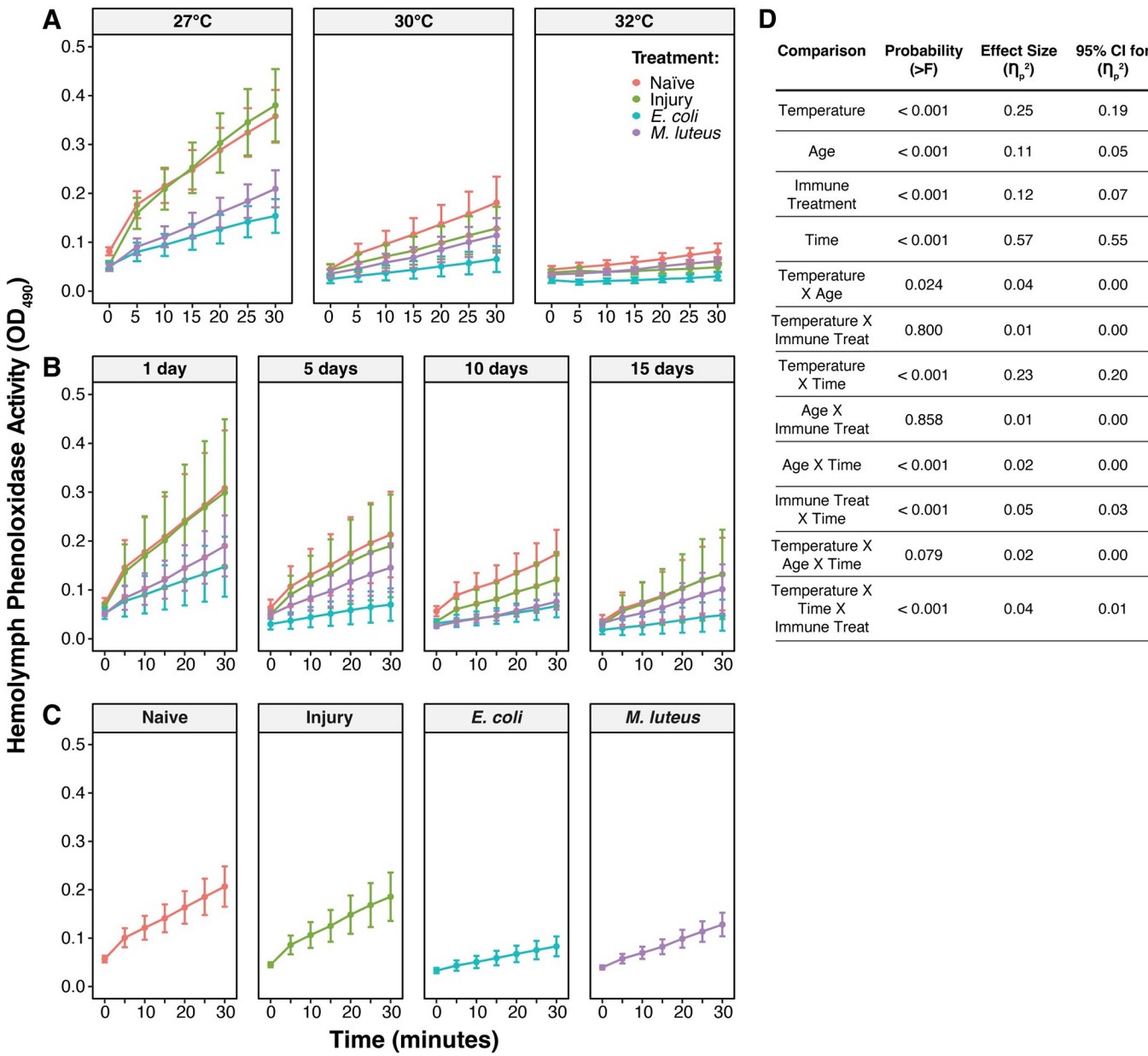

**Fig 5. Over time, melanization decreases with aging, higher temperature, and infection. A.** Melanization activity over time, aggregated by temperature and immune treatment, irrespective of age. **B.** Melanization activity over time, aggregated by age and immune treatment, irrespective of temperature. **C.** Melanization activity over time, aggregated by immune treatment, irrespective of temperature or age. Each circle marks the estimated marginal mean, and whiskers indicate the S.E.M. **D.** Statistical analyses of the data using a linear mixed-effects model and a Type II Wald Chi Square Test. The same measurements are plotted in Figs 5 and 6, but grouped or arranged differently, with aggregated data shown in this figure.

Temperature and age interact to reduce the speed of melanization more rapidly, accounting for 4% of the variation (Figs 5D and 6). Moreover, the reduction in PO activity over time with higher temperature was more pronounced in infected mosquitoes. Altogether, these interactions indicate that the shape of the increase in hemolymph melanization over the course of the assay is governed by the combined effects of higher temperature, older age, and infection. Mainly, the increase in PO activity over time is smallest in older mosquitoes that are at higher temperatures. Other interactions did not meaningfully affect PO activity, such as the

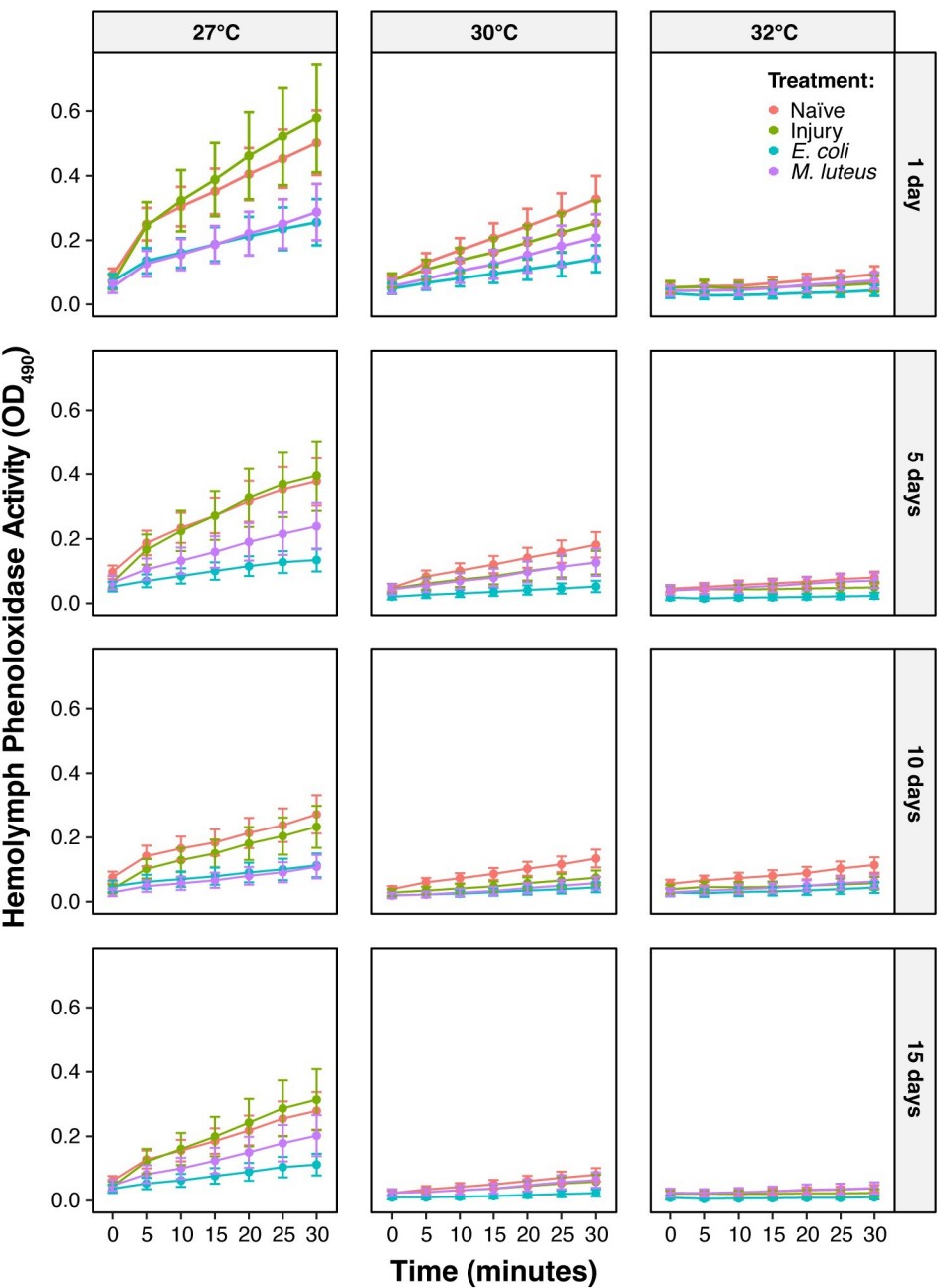

**Fig 6. Over time, the aging-dependent decline in melanization is accelerated by higher temperature.** Each circle marks the estimated marginal mean, and whiskers indicate the S.E.M. The same measurements are shown in Figs 5 and 6, but grouped or arranged differently, with unaggregated data shown in this figure. The outcomes of statistical analyses are presented in Fig 5D.

interaction between immune treatment and age, the interaction between immune treatment and temperature, and the three-way interaction between time, age, and temperature (Fig 5D).

In summary, the speed of melanization significantly decreases with higher temperature, aging, and infection. Furthermore, higher temperature accelerates the aging-dependent deterioration of the melanization immune response.

## Melanin deposition on the dorsal abdominal wall decreases with aging, and this decrease is accelerated by higher temperature

Microorganisms that enter the hemocoel circulate with the hemolymph, where they are often melanized. Melanized microorganisms are subsequently phagocytosed by hemocytes that circulate with the hemolymph or are attached to tissues [22,59–61]. Most attached hemocytes, called sessile hemocytes, are attached to the dorsal abdominal wall [53], and therefore, we set out to determine whether higher temperature, aging, and their interaction affect the melanization of pathogens by examining melanin deposition on the dorsal abdominal wall (Fig 2).

Melanin deposition on the dorsal abdominal wall is not strongly shaped by temperature, with this variable only accounting for 4% of the variation (Fig 7A and 7D). Melanin deposition is greatest at 30°C; relative to mosquitoes at 27°C, melanin deposition was 25% and 8% greater at 30°C and 32°C, respectively (Fig 7A and 7D).

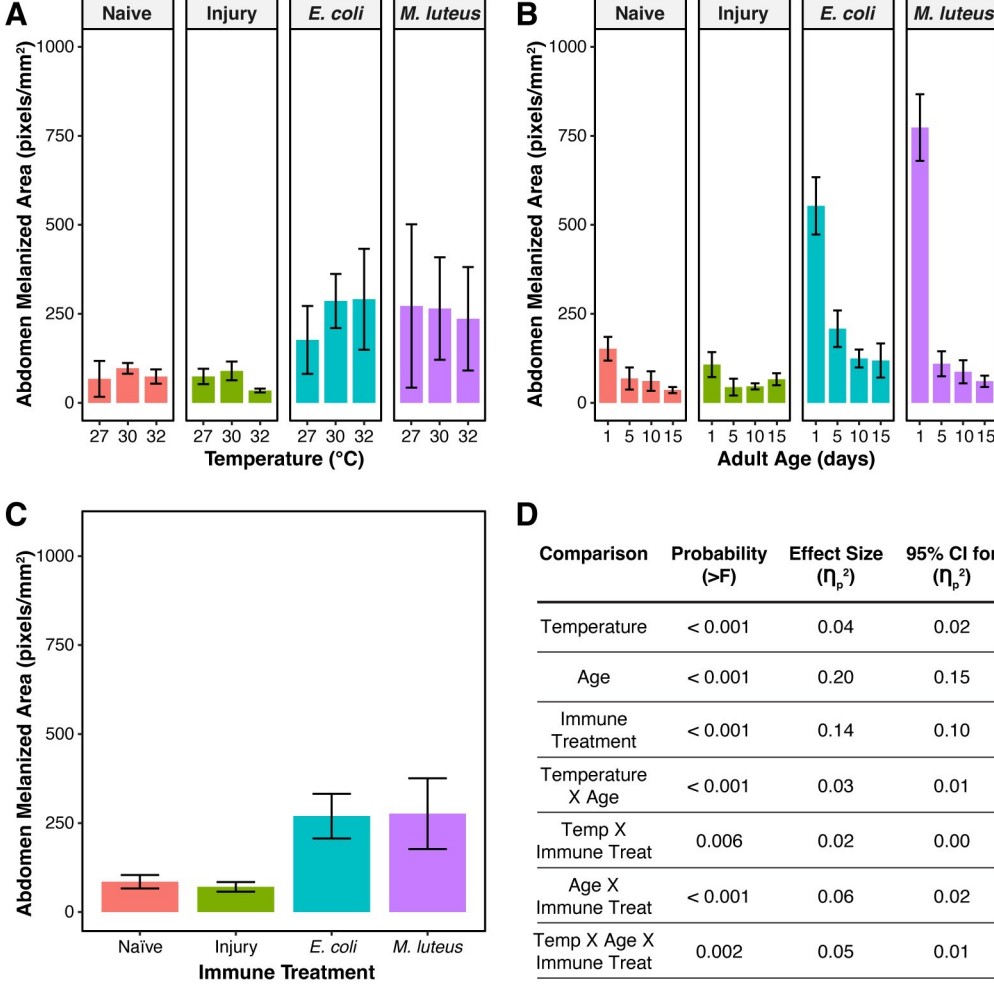

**Fig 7. Melanin deposition on the dorsal abdominal wall decreases with aging. A.** Melanin deposition, aggregated by temperature and immune treatment, irrespective of age. **B.** Melanin deposition, aggregated by age and immune treatment, irrespective of temperature. **C.** Melanin deposition, aggregated by immune treatment, irrespective of temperature or age. Column height marks the estimated marginal mean, and whiskers indicate the S.E.M. **D.** Statistical analyses of the data using a linear model and a Type II ANOVA. The same measurements are plotted in Figs 7 and 8, but grouped or arranged differently, with aggregated data shown in this figure.

Melanin deposition on the dorsal abdominal wall is predominantly shaped by aging, and this accounted for 20% of the variation (Fig 7B and 7D). Melanin deposition decreased with aging, with the greatest decrease occurring between 1 and 5 days post eclosion. Specifically, relative to 1-day-old mosquitoes, melanin deposition in 5-, 10-, and 15-day-old mosquitoes was 73%, 80%, and 82% lower, respectively (Fig 7).

With respect to immune treatment, melanin deposition on the dorsal abdominal wall of naïve and injured mosquitoes is negligible because these mosquitoes are not actively melanizing pathogens. This was expected given prior observations [59,60,62]. Because infection induces melanization, infection for 24 h increases melanin deposition, and this accounted for 14% of the variation (Fig 7C and 7D). Specifically, the melanized area of *E. coli-* and *M. luteus-* infected mosquitoes was 217% and 225% greater, respectively, than the melanized area in naïve mosquitoes.

Although higher temperature alone did not significantly affect melanin deposition on the dorsal abdominal wall, the aging-dependent decrease in melanin deposition accelerated when the temperature was higher, with the interaction between temperature and age accounting for 3% of the variation (Figs 7D and 8). As an example using *E. coli*-infected mosquitoes only, the melanized area at 27˚C decreased by 70% when mosquitoes aged from 1 to 5 days old, but at 32˚C, this aging-based decrease was 75%, indicating a larger decline when the temperature is higher.

Furthermore, temperature and immune treatment interact to shape melanin deposition, and this accounted for 2% of the variation (Figs 7D and 8). This interaction is driven by the

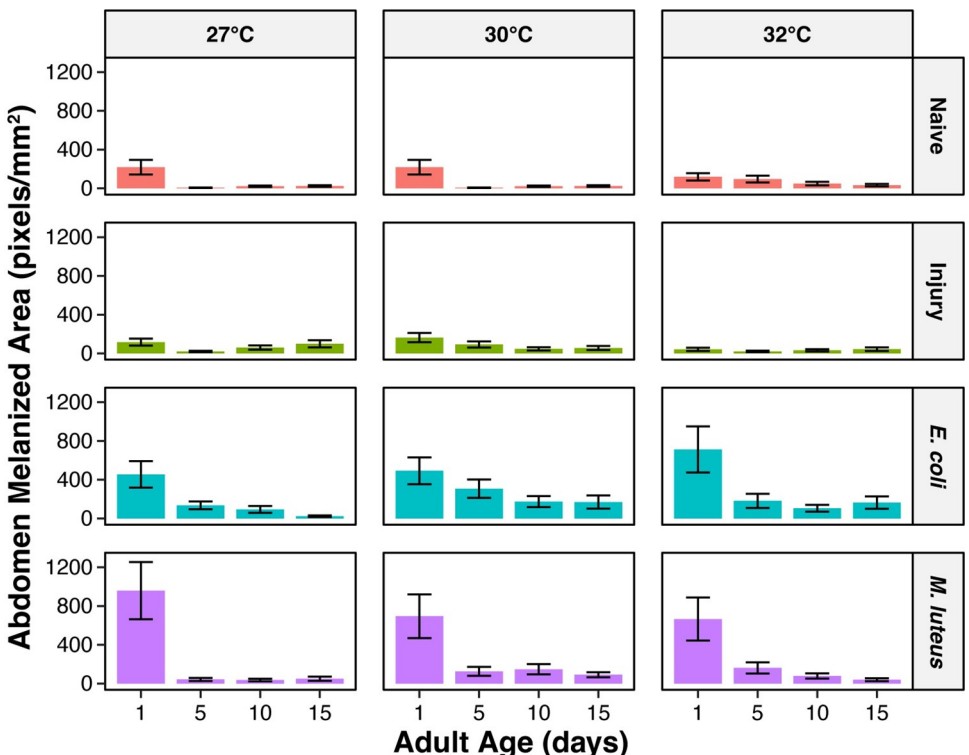

**Fig 8. The aging-associated decrease in melanin deposition is accelerated by higher temperature.** Column height marks the estimated marginal mean, and whiskers indicate the S.E.M. The same measurements are shown in Figs 7 and 8, but grouped or arranged differently, with unaggregated data shown in this figure. The outcomes of statistical analyses are presented in Fig 7D.

presence or absence of infection. In the absence of infection, the amount of melanin deposition at all temperatures is similar because pathogens are not being melanized. However, when a mosquito is infected, the effect of higher temperature depends on the pathogen. Specifically, higher temperature induced a decrease in melanin deposition following infection with *M. luteus*, but an increase following infection with *E. coli* (Fig 7A). For example, in 1-day-old mosquitoes, as temperatures warmed from 27˚C to 32˚C, the melanized area decreased by 31% in *M. luteus*-infected mosquitoes but increased by 57% in *E. coli*-infected mosquitoes.

The interaction between age and immune treatment also shapes melanin deposition, and this interaction accounted for 6% of the variation (Figs 7D and 8). Specifically, the aging-dependent decline in melanin deposition was only meaningful after infection, and the aging-dependent decline was more pronounced in *M. luteus*-infected mosquitoes than in *E. coli*-infected mosquitoes. For example, at 30˚C, as mosquitoes aged from 1 to 15 days old, melanin deposition decreased by 87% and by 66% when infected with *M. luteus* and *E. coli*, respectively.

Finally, temperature, age, and immune treatment interact to shape melanin deposition, and this three-way interaction accounted for 5% of the variation (Figs 7D and 8). Similar to the two-way interactions, this three-way interaction is driven by the infection status because in uninfected mosquitoes, the deposition of melanin is similar at all temperatures and ages because pathogens are not being melanized. As an example using *M. luteus*-infected mosquitoes only, and relative to 1-day-olds maintained at 27˚C, 1-day-olds maintained at 32˚C had 31% lower melanin deposition. As mosquitoes aged from 1 to 5 days old, the melanized area in mosquitoes at 27˚C decreased by 96%, but the melanized area in mosquitoes at 32˚C decreased by a smaller 76%.

In summary, melanin is deposited on the dorsal abdominal wall after an infection, but the amount of melanin deposited decreases when the infection is initiated at an older age. Furthermore, higher temperature accelerates the aging-dependent decline in melanization.

## Melanin deposition within the periostial regions decreases with aging but is only marginally affected by higher temperature

Within the dorsal abdominal wall, an infection for 24 h induces the aggregation of hemocytes around heart valves called ostia [61]. These hemocytes are called periostial hemocytes, and they reside in the periostial regions of abdominal segments 2–7 [60]. Abdominal segment 8 contains the posterior excurrent opening, and few sessile hemocytes reside in this segment [63,64]. Because the number of hemocytes, including periostial hemocytes, decreases with aging [42,48,53,54], we hypothesized that melanin deposition in the periostial regions decreases with aging and higher temperature, and that the effect of aging is amplified at higher temperature. To test this hypothesis, we reanalyzed the same abdomens sampled above but restricted the analysis to the periostial regions in abdominal segments 3–7 and the posterior excurrent opening.

Although influential for the entire abdomen, temperature alone did not meaningfully affect melanin deposition in the periostial regions, accounting for none (0%) of the variation (Fig 9A and 9D). Aging, however, decreased melanin deposition within the periostial regions, with age accounting for 10% of the variation (Fig 9B and 9D). Like for the entire dorsal abdominal wall, the aging-related decrease in melanization was small when mosquitoes aged beyond 5 days. Immune treatment accounted for 26% of the variation (Fig 9C and 9D); melanin deposition was negligible in uninfected and injured mosquitoes but increased dramatically after infection. Relative to naïve mosquitoes, melanin deposition was 13 times greater in *E. coli*-infected mosquitoes and 44 times greater in *M. luteus*-infected mosquitoes.

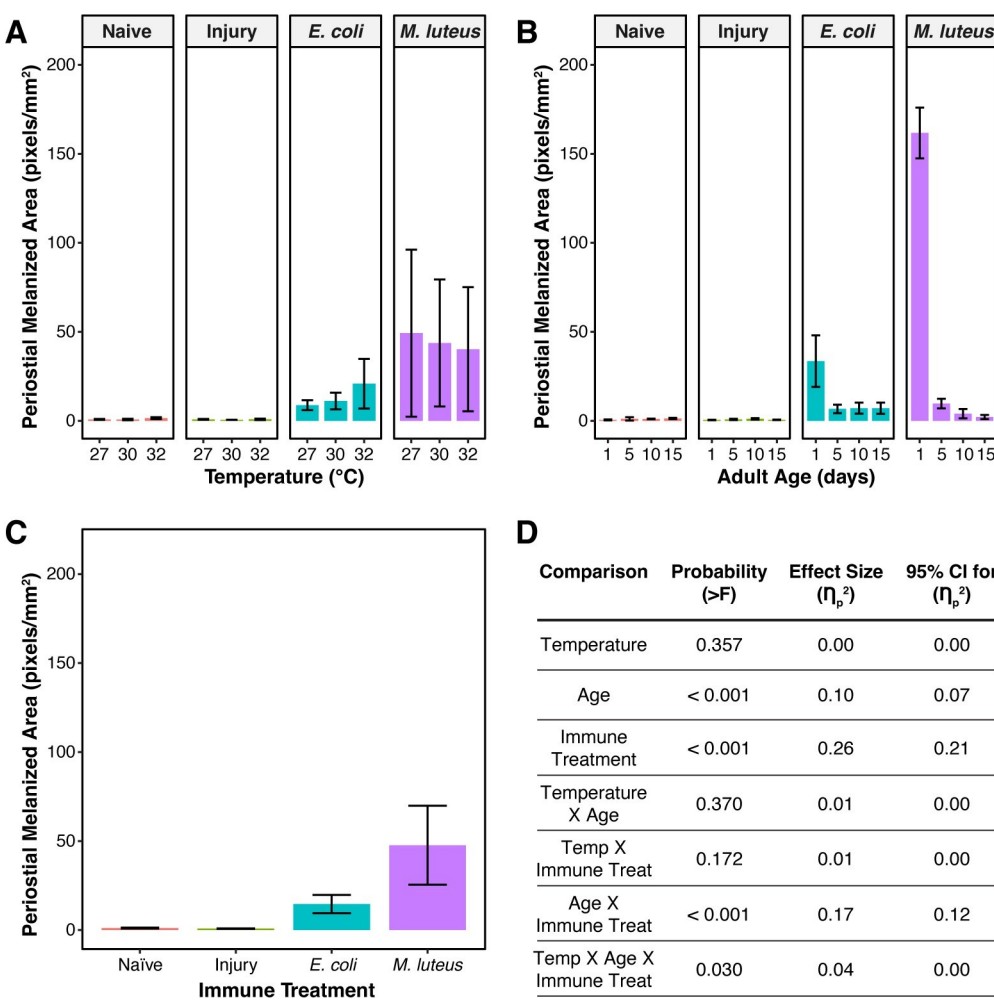

**Fig 9. Melanin deposition within the periostial regions and posterior excurrent opening decreases with aging but is only marginally affected by higher temperature. A.** Melanin deposition, aggregated by temperature and immune treatment, irrespective of age. **B.** Melanin deposition, aggregated by age and immune treatment, irrespective of temperature. **C.** Melanin deposition, aggregated by immune treatment, irrespective of temperature or age. Column height marks the estimated marginal mean, and whiskers indicate the S.E.M. **D.** Statistical analyses of the data using a linear model and a Type II ANOVA. The same measurements are plotted in Figs 9 and 10, but grouped or arranged differently, with aggregated data shown in this figure.

When examining interactions, the aging-associated decrease in melanization was dependent on the infection status, with this interaction accounting for 17% of the variation (Figs 9D and 10). Of interest, the aging-associated decrease in melanization was more pronounced in *M. luteus*-infected mosquitoes than in *E. coli*-infected mosquitoes. As an example, at 27˚C the melanized area in *M. luteus*-infected mosquitoes decreased by 99.8% between 1 and 15 days after eclosion, but it only decreased by 84% in *E. coli*-infected mosquitoes. This is likely because in 1-day-old mosquitoes, the melanization of *M. luteus* is much stronger than the melanization of *E. coli*.

Although temperature did not independently affect melanin deposition within the periostial regions, temperature interacted with age and infection status, with this three-way interaction accounting for 4% of the variation (Figs 9D and 10). Similar to the whole abdomen, this interaction was driven by the presence or absence of infection. Without infection, melanization was negligible. In infected mosquitoes, however, the aging-dependent reduction in

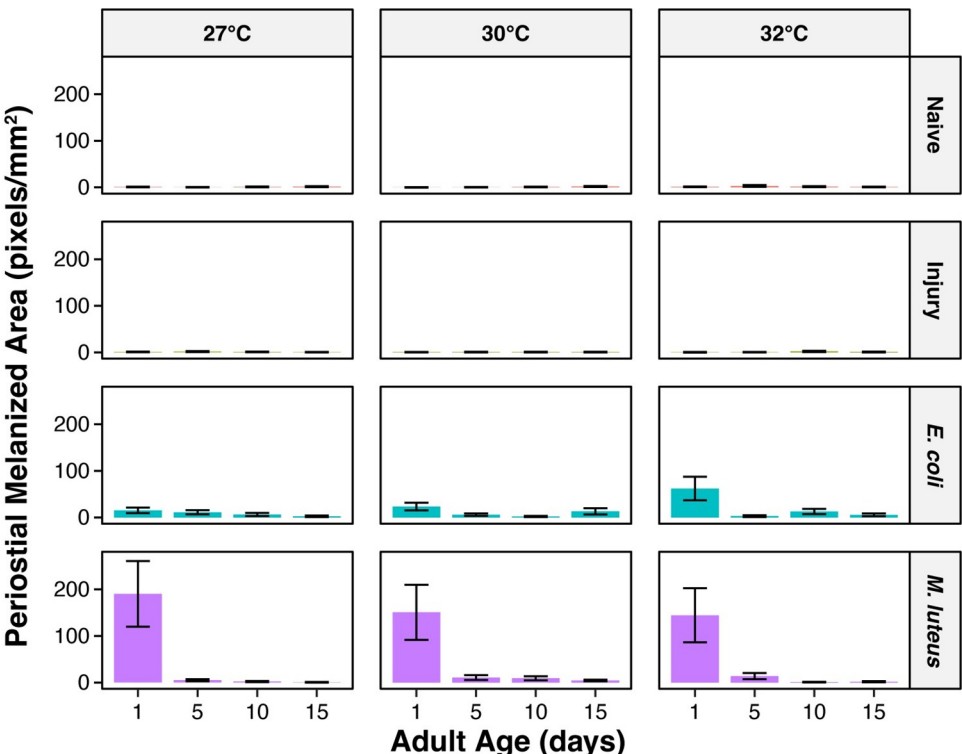

**Fig 10. The aging-associated decrease in melanin deposition within the periostial regions and posterior excurrent opening is only marginally accelerated by higher temperature.** Column height marks the estimated marginal mean, and whiskers indicate the S.E.M. The same measurements are shown in Figs 9 and 10, but grouped or arranged differently, with unaggregated data shown in this figure. The outcomes of statistical analyses are presented in Fig 9D.

melanin deposition was accelerated when the temperature was higher. This effect was more pronounced when mosquitoes were infected with *M. luteus*, likely because this infection induces the strongest melanization response. Additionally, similar to the entire dorsal abdominal wall, higher temperature had opposing effects depending on the pathogen, inducing a decrease in melanin deposition following infection with *M. luteus*, but an increase following infection with *E. coli* (Fig 9A).

In summary, melanin is deposited within the periostial regions only after infection. In infected mosquitoes, melanin deposition within the periostial region decreases with aging. Moreover, the aging-dependent decline in melanin deposition was more pronounced when the temperature was higher, especially in *M. luteus*-infected mosquitoes.

## Discussion

Although the independent effects of higher temperature and aging on the mosquito immune system have been studied, it remained unknown whether temperature and age interact to shape the immune response in mosquitoes or any other insect. We found that higher temperature and aging individually reduce melanization, and that higher temperature accelerates the aging-induced weakening of the melanization immune response (Fig 11).

Using both an *ex vivo* biochemical assay and an *in vivo* optical assay, we showed that the strength of the melanization response decreases with aging. This aging-dependent decline is in agreement with prior studies on mosquitoes [47–52]. For example, *Culex pipiens* (order:

| | Melanization in the hemolymph | | Melanin deposition on the abdominal wall | |
|---|---|---|---|---|
| | **Melanization potential** | **PO activity over time** | **Dorsal abdomen** | **Periostial regions** |
| **Does higher temperature reduce melanization?** | Yes | Yes | Yes (in *M. luteus*-infected mosquitoes) | Yes (in *M. luteus*-infected mosquitoes) |
| **Does aging reduce melanization?** | Yes | Yes | Yes (in infected mosquitoes) | Yes (in infected mosquitoes) |
| **Does higher temperature accelerate senescence?** | Yes | Yes | Yes (in infected mosquitoes) | Yes (in infected mosquitoes) |

**Fig 11. Summary of the effects of higher temperature, aging, and their interaction on the melanization immune response.** Graphic that includes cuvettes created with BioRender.com.

Culicidae) loses over half of their PO activity by 14 days of adulthood [50], and aging reduces PO activity in crickets (order: Orthoptera) [65]. This aging-related decrease in melanization correlates with a decrease in the number of hemocytes [42,48,53,54], which is important because hemocytes—mainly oenocytoids but also to some extent granulocytes—are major producers of key melanization enzymes such as phenoloxidase and phenylalanine hydroxylase [23,54,66,67].

Unexpectedly, aging beyond 10 days did not meaningfully alter the mosquito's melanization immune response. In insects, melanin is not only necessary for immune function, but it is also required for the sclerotization and pigmentation of the cuticle during molting [56,68,69]. Given that melanization is energetically costly [56,70], once the mosquito has reached adulthood and the cuticle has fully hardened, we suspect that there is a lower need for PO activity, and therefore, resources are deployed elsewhere. This is supported by mosquito larvae having much greater PO activity than adults [51], and similar to our findings, the aging-based weakening of the melanization of microfilaria is marginal beyond 5 days of age [48]. Thus, the plateau in the aging-associated decline of melanization is likely a consequence of achieving an optimal investment in melanization potential during adulthood.

We also discovered that higher temperature weakens the melanization immune response. Similar to our findings, the efficiency of C-type lectin 4-regulated melanization of *Plasmodium falciparum* in *A. gambiae* is less efficient when the temperature warms from 19°C to 27°C [20]. Moreover, higher temperature reduces the melanization of Sephadex beads that have been injected into *A. gambiae* or *Anopheles stephensi* [15,18]. Higher temperature also reduces the strength of melanization in butterflies (order: Lepidoptera) and crickets [71–73], indicating that this phenomenon transcends the mosquito taxon. Additionally, higher temperature

reduces the sclerotization and melanization of the cuticle, resulting in lighter colored insects [74]. In moths, beetles (order: Coleoptera) and crickets, a lighter cuticle correlates with a weaker melanization immune response [75–79], and in beetles, a lighter cuticle also correlates with a lower hemocyte density [79]. Thus, higher temperature weakens both cuticular and immune melanization across Insecta.

Prior to this study, it remained unknown whether or how temperature and age interact to shape the immune response in insects. Without considering interactive effects, both higher temperature and aging reduce mosquito survival and vector competence [19,31,32,35,80]. Recently, we reported that higher temperature and aging, individually and interactively, reduce body size and deteriorate the body condition of *A. gambiae* [12]. Here, we discovered that higher temperature and aging significantly interact to accelerate the weakening of the melanization response. To our knowledge, our study provides the first evidence that higher temperature accelerates immune senescence in mosquitoes—or any other insect—thereby decoupling physiological age from chronological age. Previously, we reported that the melanization of mosquito hemolymph is inhibited by a copper-specific enzyme inhibitor, diethyl-dithiocarbamate (DETC), indicating that melanization is driven by PO [51], and we and others have demonstrated an aging-related decline in the number of hemocytes [42,48,53,54]. Therefore, we conclude that the warming-based acceleration of the aging-dependent decline in melanization is because of (i) an accelerated decline in PO availability, and (ii) an accelerated decline in the number of hemocytes that produce PO and other melanization enzymes.

The aging-dependent weakening of melanization potential is less pronounced when the temperature is higher. This is largely because mosquitoes reared at higher temperatures eclose with a lower melanization potential, which resembles that of older adults that had been reared at cooler temperatures. While this may seem to indicate a lack of aging-based weakening at higher temperatures, it is more likely that the aging-based weakening at higher temperature occurs during the immature stages and prior to adult emergence, reducing the initial adult potential for melanization. This notion is supported by the findings that larvae have greater melanization activity than adults [51], and that higher temperature reduces cuticular melanization [73,78], which is needed during eclosion [68,69].

Infection for 24 h activates the immune system and increases melanin deposition on the dorsal abdominal wall, so it may seem counterintuitive that infection for 24 h weakens the melanization potential of hemolymph. We hypothesize that the reduction we observed is due to PO enzyme depletion. PO enzymes are found in melanotic capsules following bacterial infection in *Aedes aegypti* [67], and upon wounding, PO enzymes also localize to the cuticle healing sites in *Armigeres subalbatus* [58]. Moreover, the expression of PO genes is not significantly upregulated following a hemocoelic infection in adult mosquitoes [51], and without a blood-meal, the total protein content of a mosquito decreases with older age [12,49]. Therefore, we suspect that the availability of PO in the hemolymph declines as PO is sequestered in the sites of bacterial infection or wounding, reducing the remaining melanization potential. In our spectrophotometric experiments, we provided a saturating amount of PO's substrate, L-DOPA, but inside the mosquito, it is likely that substrate reduction also contributes to reduced melanization potential upon infection. This notion is supported by a study where filarial worms were injected into the hemocoel of *A. subalbatus*, which found that the melanization response is accompanied by a reduction in tyrosine, one of the initial substrates of the melanization cascade [81]. Regardless of the specific reason for the reduction in melanization potential at 24 h following infection, the melanization potential in naïve mosquitoes represents the melanization activity that can take place at the initiation of any infection. The melanization potential in mosquitoes infected for 24 h then informs on how the progression of an infection affects this immune response.

Although infection for 24 h decreases melanization potential, it increases melanin deposition. This is because infection activates the melanization response, leading to the phagocytosis and sequestration of melanized bacteria by sessile hemocytes on the abdominal wall [22,53,59–61]. Interestingly, melanin deposition in mosquitoes infected with *E. coli* was greatest at the highest temperature, even when melanization potential was not. This may at first suggest a beneficial effect of higher temperature, but we do not believe that this is the case. Instead, we hypothesize that because the highest temperature is closer to the optimal growth temperature of *E. coli*, bacterial proliferation increases, leading to a greater melanization response. The optimal temperature for the growth of *M. luteus* is lower than for *E. coli*, and hence, this rising temperature trend is not observed for *M. luteus*.

With infection, the robustness of the melanization response was greater in *M. luteus*-infected mosquitoes than in *E. coli*-infected mosquitoes, and the aging-dependent decline in melanization was more pronounced in *M. luteus*-infected mosquitoes. Different types of bacteria preferentially activate phagocytosis versus melanization responses [82]. In *A. subalbatus* and *A. aegypti*, *E. coli* are primarily phagocytosed by hemocytes, whereas *M. luteus* are primarily melanized [22,23]. Given that the weakening of melanization was more pronounced in *M. luteus*-infected mosquitoes, we conclude that the warming-based acceleration of the aging-dependent decline of melanization is more pronounced when melanization is the major immune response elicited by the pathogen.

The experiments presented here were all conducted in the absence of a blood meal, and it is possible that the absence of a blood meal exacerbates the senescence of the melanization response. Upon ingesting a bloodmeal, mosquitoes synthesize cholesterol and convert it into the hormone 20-hydroxyecdysone (20E), which is a pleiotropic steroid that regulates molting, development, immunity, and longevity [83–85]. In *A. gambiae*, 20E primes and strengthens the immune response against *P. falciparum* [86,87], including PO production [85,88]. However, given that the mosquitoes in our study did not ingest a bloodmeal, we hypothesize that older adults may have less 20E, and therefore, weaker activation of hormone-regulated immunity. Moreover, a blood meal induces hemocyte proliferation [89,90], and this opens the possibility of an enhanced melanization response. Therefore, future studies will incorporate blood feeding in the presence and absence of infection to test its effects on the senescence of the melanization immune response.

This study did not consider mosquito survival as part of the analysis. However, the risk of death increases with aging [36,41,42], higher temperature shortens lifespans [7,9,14], and infection decreases survival [42,82]. How age, temperature and infection interact to shape mosquito survival is the focus of ongoing experiments in our laboratory, but the rate of mosquito death does not affect our conclusions on melanization. Here, we only assessed melanization in the surviving mosquitoes because these are the only mosquitoes that, under the conditions tested, would be available for pathogen acquisition and transmission.

In summary, this study demonstrates that higher temperature accelerates immune senescence in mosquitoes, with higher temperature uncoupling physiological age from chronological age (Fig 11). These findings, which we believe are the first to show this interactive effect in any insect, highlight the need to holistically examine the impact of warming global temperatures on the ability of insects to transmit diseases to humans, animals and plants, or their ability to serve as pollinators for our food supply. Given that many abiotic and biotic factors—such as temperature and age—alter mosquito physiology and vector competence [91], the present study illuminates the importance of accounting for the interactive effects of the mosquito's environment on internal physiology, which is inherently important when estimating immune function, disease transmission dynamics, and other critical processes.

## Materials and methods

### Mosquito rearing, treatments, and experimental overview

A laboratory colony of *Anopheles gambiae*, Giles *sensu stricto* (G3 strain; Diptera: Culicidae) was maintained at 27°C, 75% relative humidity, and a 12h:12h light:dark photoperiod. Eggs from this colony were collected and transferred to three environmental chambers held at 27°C, 30°C or 32°C, where they were hatched and reared to adulthood, and then used for experimentation. These temperatures were selected because they are experienced by *A. gambiae* in nature and represent warming global temperatures [1,92,93]. Larvae were fed a mixture of 2.8 parts koi food to 1 part baker's yeast daily, and pupae were separated daily. Upon eclosion, adults were maintained in 2.4 L plastic buckets with a mesh marquisette top and were fed 10% sucrose solution ad libitum.

For all experiments, adult females at 1, 5, 10, and 15 days after eclosion were assessed in each temperature (Fig 1). These ages were selected because substantial changes in mosquito immunity occur with adult aging, and these ages encompass the timeline for *Plasmodium* parasite development within this mosquito [30,42,47,94].

At each temperature and age, adult female mosquitoes were either (i) naïve (unmanipulated), (ii) injured, (iii) infected with *Escherichia coli* (Gram-negative bacteria; modified DH5α, GFP-expressing and tetracycline resistant), or (iv) infected with *Micrococcus luteus* (Gram-positive bacteria; ATCC 4698). *E. coli* and *M. luteus* were grown overnight in Luria-Bertani (LB) broth at 37°C in a shaking incubator (New Brunswick Scientific, Edison, NJ, USA), and the cultures were then normalized to $OD_{600} = 2$. Mosquitoes were anesthetized on ice and injected into the hemocoel with 69 nL of sterile LB (injured mosquitoes) or a bacterial culture (infected mosquitoes) in the thoracic anepisternal cleft, using a Nanoject III Programmable Nanoliter Injector (Drummond Scientific Company, Broomall, PA, USA). Absolute infection doses were determined by diluting the cultures, plating them on LB + tetracycline agar plates (*E. coli*) or LB-only agar plates (*M. luteus*), and counting the colony forming units (CFUs) that grew. Across experimental trials, the infection doses averaged at 12,203 *E. coli* per mosquito and 6,865 *M. luteus* per mosquito. Throughout this study, the age of the mosquito represents the age when the immune treatment was initiated.

### Quantification of phenoloxidase activity in hemolymph

To quantify the melanization potential of hemolymph, or the ability of hemolymph to melanize a pathogen upon infection using active PO, we used a spectrophotometric assay as previously described [51, 95]. For each condition, hemolymph was extracted from 18–25 mosquitoes at 24 h after the immune treatment. Briefly, the lateral thorax of cold-anesthetized females was punctured using a 0.20 mm diameter minutien insect pin and the mosquitoes were placed inside a 0.6 mL microfuge tube that contained a small incision at the bottom. The 0.6 mL tube was nested inside a 1.5 mL microfuge tube and then centrifuged at 5000 RCF for 5 min at 4°C. The ~1–2 μL of hemolymph that pooled at the bottom of the 1.5 mL tube was collected and stored at -20°C until further use.

The PO activity of hemolymph was quantified using a biochemical assay that measures the conversion of 3,4-Dihydroxy-L-phenylalanine (L-DOPA, clear, $\lambda_{max}$ of 280 nm) to dopachrome (reddish-brown, $\lambda_{max}$ of 475 nm) [51,96,97] (Fig 2A and 2B). For this assay, 1 μL of hemolymph was added to 50 μL of deionized water. Then, 10 μL of that mixture was added to a cuvette containing 90 μL of 4 mg/mL L-DOPA (Sigma, St. Louis, MO, USA), and the absorbance at $OD_{490}$ was measured every 5 min for 30 min using a BioPhotometer Plus spectrophotometer (Eppendorf AG, Hamburg, Germany). On average, 7 independent biological trials

were conducted for each temperature-age-immune treatment combination. In total, 317 hemolymph samples, derived from approximately 6800 mosquitoes, were assayed. A negative control was conducted during each trial, where 10 μL of water was added to a cuvette containing 90 μL of 4 mg/mL L-DOPA. Additional controls were conducted using a subset of samples, in which diluted hemolymph was added to 90 μL of water.

## Quantification of melanization on the dorsal abdominal wall

To measure the melanization that occurs inside a mosquito, we imaged the dorsal abdominal wall and quantified the dark melanin deposits using a light intensity method previously described [51,59,60]. For each condition, mosquitoes were cold-anesthetized at 24 h after immune treatment and fixed by injecting cold, 16% paraformaldehyde (Electron Microscopy Sciences, Hatfield, PA) into the hemocoel. After 10 min on ice, abdomens were bisected along the coronal plane and immersed in PBS containing 0.1% Triton X-100. Internal organs were removed, and the dorsal abdomens were rinsed in PBS and mounted flat on glass slides with coverslips using Aqua-Poly/Mount (Polysciences, Warrington, PA, USA).

Dorsal abdominal segments 3–8 were imaged under brightfield illumination using a 10x objective on a Nikon Eclipse Ni-E compound microscope connected to a Nikon Digital Sight DS-Qi1 monochrome digital camera and Nikon Advanced Research NIS Elements software (Nikon, Tokyo, Japan). For each dorsal abdomen, two Z-stacks were captured using a linear encoded Z-motor. The first stack contained abdominal segments 3–5 whereas the second stack contained segments 6–8. Each Z-stack was then rendered into a focused, two-dimensional image using the Extended Depth of Focus (EDF) function in NIS Elements (Fig 2C).

For each mosquito, each abdominal segment was delineated using the region of interest (ROI) tool in NIS Elements, and each periostial region (the region surrounding the heart valves in segments 3–7) and the excurrent opening (in segment 8) was further delineated. Then, an intensity threshold was set such that it distinguished melanized areas (pixels below the threshold) from non-melanized areas (pixels above the threshold). Three independent biological trials were conducted for each temperature-age-immune treatment combination, with a minimum of three mosquitoes per group. On average, each combination contained 16 mosquitoes assayed across 3 independent biological trials, with 760 mosquitoes assayed in total.

## Statistical analysis

Statistical analyses were completed using R Statistical Software, v4.2.2 [98]. The spectrophotometric data were analyzed in two ways: (i) using the $OD_{490}$ reading at 30 min as a measure of the final melanization potential, and (ii) using the change in $OD_{490}$ over the course of the 30 min experiment as a measure of PO activity over time. Data for each temperature-age-immune treatment combination were first tested for normality using the Shapiro-Wilk test and were found to be non-normal. Thus, the data were zero-adjusted and log-transformed to achieve normality.

For data on the final melanization potential, we used a linear model to identify the main effects of temperature, age, immune treatment, and the interaction between temperature and age. Other interactions (e.g., *temperature x immune treatment*) did not meaningfully contribute to melanization potential, so they were excluded from the model.

For data on PO activity over time, we used a linear mixed-effects model, fit by maximum likelihood, to identify the relationship between $OD_{490}$ and the main effects of time, immune treatment, age, and temperature using the "lme4" package [99]. Two-way interactions among the main effects (*time x immune treatment*; *time x age*; *time x temperature*; *immune treatment x age*; *immune treatment x temperature*; *age x temperature*), three-way interactions among the

main effects (*time x immune treatment x temperature*; *time x age x temperature*), and the random effect of the individual hemolymph sample were included as predictors in the model.

Melanin deposition on the dorsal abdominal wall was analyzed the same way for two sets of data: (i) the melanized area on the entirety of dorsal abdominal segments 3–8, and (ii) the melanized area within the periostial regions in segments 3–7 plus the excurrent opening of segment 8. Data were non-normal, so they were zero-adjusted and log-transformed to achieve normality. Then, we used a linear model to identify the main effects of higher temperature, aging, infection, and their interactions (*temperature x age*; *temperature x immune treatment*; *age x immune treatment*; *temperature x age x immune treatment*) on the melanized area.

For all statistical analyses, final models were determined by a stepwise, multidirectional selection method, comparing model residuals, log-likelihood ratios, and Akaike Information Criterion (AIC) values. We then conducted type-II ANOVAs with Wald Chi Square Tests on the final models using the "car" package [100]. Partial effect sizes and 95% confidence intervals were calculated using the "effectsize" package [101]. To assess the effects of the interaction between temperature and age, for each statistical model we calculated estimated marginal means using the "emmeans" package on the response scale, also known as "least-square means" [102,103]. Sidak-adjusted pairwise contrasts of the estimated marginal means were then performed within each temperature-age-immune treatment combination to identify significant differences between groups.

Estimated marginal means, ANOVA and Chi Square p-values, partial effect sizes, and effect size confidence intervals are presented in the main figures. Observed means for main melanization response variables and controls are presented in S1–S10 Figs. Additional information is presented in the supplement, including the raw data, raw means, estimated marginal means, model coefficients, and full ANOVA and Chi Square tables (S1–S3 Files).

Figures depicting methods were created with BioRender.com, whereas graphs depicting data were created with R and assembled into figures using Adobe Illustrator. Graphics created using BioRender.com are published under agreement numbers WV25ZSF5QR (Fig 1), XC25ZSFWGC (Fig 2), and WG25ZSGDUE (Fig 11).

## Supporting information

**S1 Fig. Auto-oxidation of exogenous L-DOPA is negligible.** Time course of $OD_{490}$ measurements of L-DOPA plus water for 30 min. The lower end of the scale is amplified on the right. No meaningful auto-oxidation of L-DOPA was detected. Each circle marks the mean, and whiskers indicate the S.E.M.
(PDF)

**S2 Fig. Melanization in isolated hemolymph is negligible without the addition of the phenoloxidase substrate, L-DOPA.** Time course of $OD_{490}$ measurements of hemolymph for 30 min. The lower end of the scale is amplified on the right. No meaningful melanization was detected in the absence of exogenous L-DOPA. Each circle marks the mean, and whiskers indicate the S.E.M.
(PDF)

**S3 Fig. Raw means of melanization potential, aggregated by temperature, age, and immune treatment. A.** Melanization potential, aggregated by temperature and immune treatment, irrespective of age. **B.** Melanization potential, aggregated by age and immune treatment, irrespective of temperature. **C.** Melanization potential, aggregated by immune treatment, irrespective of temperature or age. Column height marks the raw mean, and whiskers indicate the S.E.M. The same measurements are plotted in S3 and S4 Figs, but grouped or arranged differently,

with aggregated data shown in this figure. The estimated marginal means of these data, resulting from the linear model, are presented in Fig 3.
(PDF)

**S4 Fig. Raw means of melanization potential.** Column height marks the raw mean, and whiskers indicate the S.E.M. The same measurements are plotted in S3 and S4 Figs, but grouped or arranged differently, with unaggregated data shown in this figure. The estimated marginal means of these data, resulting from the linear model, are presented in Fig 4.
(PDF)

**S5 Fig. Raw means of melanization over time, aggregated by temperature, age, and immune treatment. A.** Melanization activity over time, aggregated by temperature and immune treatment, irrespective of age. **B.** Melanization activity over time, aggregated by age and immune treatment, irrespective of temperature. **C.** Melanization activity over time, aggregated by immune treatment, irrespective of temperature or age. Each circle marks the raw mean, and whiskers indicate the S.E.M. The same measurements are plotted in S5 and S6 Figs, but grouped or arranged differently, with aggregated data shown this figure. The estimated marginal means of these data, resulting from the linear mixed model, are presented in Fig 5.
(PDF)

**S6 Fig. Raw means of melanization over time.** Each circle marks the raw mean, and whiskers indicate the S.E.M. The same measurements are plotted in S5 and S6 Figs, but grouped or arranged differently, with unaggregated data shown in this figure. The estimated marginal means of these data, resulting from the linear mixed model, are presented in Fig 6.
(PDF)

**S7 Fig. Raw means of melanin deposition on the dorsal abdominal wall, aggregated by temperature, age, and immune treatment. A.** Melanin deposition, aggregated by temperature and immune treatment, irrespective of age. **B.** Melanin deposition, aggregated by age and immune treatment, irrespective of temperature. **C.** Melanin deposition, aggregated by immune treatment, irrespective of temperature or age. Column height marks the raw mean, and whiskers indicate the S.E.M. The same measurements are plotted in S7 and S8 Figs, but grouped or arranged differently, with aggregated data shown in this figure. The estimated marginal means of these data, resulting from the linear model, are presented in Fig 7.
(PDF)

**S8 Fig. Raw means of melanin deposition on the dorsal abdominal wall.** Column height marks the raw mean, and whiskers indicate the S.E.M. The same measurements are plotted in S7 and S8 Figs, but grouped or arranged differently, with unaggregated data shown in this figure. The estimated marginal means of these data, resulting from the linear model, are presented in Fig 8.
(PDF)

**S9 Fig. Raw means of melanin deposition within the periostial regions and posterior excurrent opening, aggregated by temperature, age, and immune treatment. A.** Melanin deposition, aggregated by temperature and immune treatment, irrespective of age. **B.** Melanin deposition, aggregated by age and immune treatment, irrespective of temperature. **C.** Melanin deposition, aggregated by immune treatment, irrespective of temperature or age. Column height marks the mean, and whiskers indicate the S.E.M. The same measurements are plotted in S9 and S10 Figs, but grouped or arranged differently, with aggregated data shown in this figure. The estimated marginal means of these data, resulting from the linear model, are presented in Fig 9.
(PDF)

**S10 Fig. Raw means of melanin deposition within the periostial regions and posterior excurrent opening.** Column height marks the mean, and whiskers indicate the S.E.M. The same measurements are shown in S9 and S10 Figs, but grouped or arranged differently, with unaggregated data shown in this figure. The estimated marginal means of these data, resulting from the linear model, are presented in Fig 10.
(PDF)

**S1 File. Data and statistical information for Figs 3–6.**
(XLSX)

**S2 File. Data for Figs 3–6 and S1–S6 and statistical information for S1–S6 Figs.**
(XLSX)

**S3 File. Data and statistical information for Figs 7–10 and S7–S10 Figs.**
(XLSX)

## Acknowledgments

We thank Drs. Courtney Murdock and Ann Tate for their helpful advice and discussion about statistical analyses. We also thank Jordyn Barr, Cole Meier, Shabbir Ahmed, Tania Estévez-Lao, and Tobias McCabe for offering comments and discussion on this manuscript.

## Author Contributions

**Conceptualization:** Lindsay E. Martin, Julián F. Hillyer.

**Data curation:** Lindsay E. Martin, Julián F. Hillyer.

**Formal analysis:** Lindsay E. Martin, Julián F. Hillyer.

**Funding acquisition:** Lindsay E. Martin, Julián F. Hillyer.

**Investigation:** Lindsay E. Martin.

**Methodology:** Lindsay E. Martin, Julián F. Hillyer.

**Project administration:** Lindsay E. Martin, Julián F. Hillyer.

**Resources:** Julián F. Hillyer.

**Software:** Lindsay E. Martin.

**Supervision:** Julián F. Hillyer.

**Validation:** Lindsay E. Martin, Julián F. Hillyer.

**Visualization:** Lindsay E. Martin, Julián F. Hillyer.

**Writing – original draft:** Lindsay E. Martin, Julián F. Hillyer.

**Writing – review & editing:** Lindsay E. Martin, Julián F. Hillyer.

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
