## [Decision Letter · Decision Letter 0]

12 Dec 2023

Dear Dr. Hillyer,

Thank you very much for submitting your manuscript "Warmer temperature accelerates the aging-dependent weakening of the melanization immune response in mosquitoes" for consideration at PLOS Pathogens. As with all papers reviewed by the journal, your manuscript was reviewed by members of the editorial board and by several independent reviewers. The reviewers appreciated the attention to an important topic. Based on the reviews, we are likely to accept this manuscript for publication, providing that you modify the manuscript according to the review recommendations.

Please address the comments that do not require additional experimental work.

Sincerely,

Elizabeth A McGraw, PhD

Academic Editor

PLOS Pathogens

Jeffrey Dvorin

Section Editor

PLOS Pathogens

Kasturi Haldar

Editor-in-Chief

PLOS Pathogens

orcid.org/0000-0001-5065-158X

Michael Malim

Editor-in-Chief

PLOS Pathogens

orcid.org/0000-0002-7699-2064

Please address the comments that do not require additional experimental work.

Reviewer Comments (if any, and for reference):

Reviewer's Responses to Questions

**Part I - Summary**

Reviewer #1: This is an interesting paper with important implications for disease resistance/tolerance and transmission in insects – and particularly in vectors. It also has interesting implications overall for host-parasite interactions in relatively short-lived insects and pressures that could shape the evolution of parasite life cycles in mosquitoes. If you develop too quickly as a parasite that needs to migrate through the hemolymph to the salivary glands, perhaps you encounter too strong an immune response; however, the timing of development could be under selective pressure to match the senescence of the immune system. It certainly has me thinking! I have mainly minor suggestions. The general execution is sound and I have really only a few suggestions for interpretation of the data.

Reviewer #2: This manuscript “Warmer temperature accelerates the aging-dependent weakening of the melanization immune response in mosquitoes” explores the interaction of increasing temperature and age on the melanization response. Laboratory strain G3 mosquitoes were reared at 3 different temperatures selected to represent a range expected from global climate change. From these conditions, mosquitoes at 4 different ages were subjected to immune challenge, injury, or control. 24 h post treatment, the melanization response was assayed using an enzymatic assay and by observation of melanin deposition in the cuticle.

Declines in melanization were observed with increased age and temperature. Furthermore, temperature speeds up the age-dependent decline. Immune challenge also, reduced melanization enzymatic activity, but increased melanotic deposits. In addition to the abdominal cuticles, melanization of the periostial region was also quantified.

**Part II – Major Issues: Key Experiments Required for Acceptance**

Reviewer #1: I suggest “higher” temperature instead of “warmer” temperature throughout. It’s finicky and not that important, but grammatically appropriate.

Thank for the experimental overview – that is very helpful!

Did any of the mosquitoes die from infection or injury? I.e. did the infection select for any particular mosquitoes?

I was wondering if a useful way to show the interaction between age and temperature would be to express active or melanization as a percent of what was available at day 1. It could show the acceleration of decline at 30 C compared to 27 C even more. On that note, I’m also curious about what is happening at 32 and perhaps noting that the interaction of aging and temperature seem null at the highest temperature. PO seems to be so low at 32 C to begin with that there really isn’t a decline with age at all; and I suspect that a 1-day-old mosquito at 32 C isn’t equivalent to 15-day-old mosquito at 30 C (i.e. they both have approx. the same PO response).

You say on line 412 that heat weakens melanization, but it does seem that in some cases the melanization response is actually fairly robust, if not stronger, at higher temps (e.g. with E. coli infection). It’s a conundrum to some extent that PO can be so low at 32 C but there can still be a strong melanization response – do you think they just don’t need as much PO at higher temps? I.e. it’s hot enough that the enzyme works quickly enough that they don’t require as much enzyme. Because they are reared at 30 C, this could represent an acclimation effects rather than an aging effect. It might be helpful to mention this in the discussion.

Reviewer #2: Overall the manuscript is clear and conclusions are generally supported by the data. Given that the effects of increasing temperature and age have independently been studied by these authors and others in different systems, the combined effect is not unexpected, and without a clear mechanism does not make a major impact for the field

The importance of this work is about mosquito vectorial capacity, so examining how age/temp/immunity are affected by blood feeding is crucial. This is acknowledged in the manuscript as something for future work, but should be included in these studies.

The lack of mechanism also detracts from the overall impact. It is suggested that for both temperature and age, hemocytes have been shown to decrease in mosquitoes or other insects, so perhaps this is the mechanism for the further decline, but it is not addressed here by hemocyte number or other mechanistic insight. There are studies that have chemically depleted hemocytes that could be employed into these analyses.

**Part III – Minor Issues: Editorial and Data Presentation Modifications**

Reviewer #1: (No Response)

Reviewer #2: It is stated that the melanization potential of the hemolymph is being measured. It is more accurate that the current melanization activity of the hemolymph is what is being measured. It would be important to challenge the age/temp ranges with bacteria acutely and then performing a melanization assay. This type of assay would more accurately reflect the potential of the hemolymph to mount a melanization response. It is not reported how many mosquitoes died following immune challenge. The assay might not accurately represent the population if the highly responding individuals die.

There are also overlapping rationales for the declines observed in the enzymatic assay. For age, senescence, presumably hemocyte decline, is cited, which is possibly accelerated with temperature. With immune challenge, the decrease is enhanced due to the consumption of the melanization components as the enzymatic assays are performed 24 h after the immune challenge. Melanotic deposits in the immune challenged mosquitoes support the immune challenge depletion model, but since both inhibition via senescence or inhibition via consumption have the same effect on the assay, the interpretation is confusing.

PLOS authors have the option to publish the peer review history of their article (what does this mean?). If published, this will include your full peer review and any attached files.

Reviewer #1: No

Reviewer #2: No

Figure Files:

Data Requirements:

Reproducibility:

References:

---

## [Editor Report · Decision Letter 1]

1 Jan 2024

Dear Dr. Hillyer,

We are pleased to inform you that your manuscript 'Higher temperature accelerates the aging-dependent weakening of the melanization immune response in mosquitoes' has been provisionally accepted for publication in PLOS Pathogens.

Best regards,

Elizabeth A McGraw, PhD

Academic Editor

PLOS Pathogens

Jeffrey Dvorin

Section Editor

PLOS Pathogens

Kasturi Haldar

Editor-in-Chief

PLOS Pathogens

orcid.org/0000-0001-5065-158X

Michael Malim

Editor-in-Chief

PLOS Pathogens

orcid.org/0000-0002-7699-2064
---

## [Editor Report · Acceptance letter]

3 Jan 2024

Dear Dr. Hillyer,

We are delighted to inform you that your manuscript, "Higher temperature accelerates the aging-dependent weakening of the melanization immune response in mosquitoes," has been formally accepted for publication in PLOS Pathogens.

Best regards,

Michael Malim

Editor-in-Chief

PLOS Pathogens

orcid.org/0000-0002-7699-2064